# Recovery of Tendon Characteristics by Inhibition of Aberrant Differentiation of Tendon-Derived Stem Cells from Degenerative Tendinopathy

**DOI:** 10.3390/ijms21082687

**Published:** 2020-04-13

**Authors:** Sun Jeong Kim, Hae Won Oh, Jong Wook Chang, Sang Jun Kim

**Affiliations:** 1Department of Physical and Rehabilitation Medicine, Stem Cell & Regenerative Medicine Institute, Samsung Medical Center, Sungkyunkwan University School of Medicine, Seoul 06351, Korea; xhxhfn78@naver.com; 2R&D Center, ENCell Co. Ltd., Seoul 06072, Korea; 3Division of Health Policy and Administration, School of Public Health, University of Illinois, Chicago, IL 60612, USA; hoh30@uic.edu; 4Department of Health Sciences and Technology, SAIHST, Sungkyunkwan University, Seoul 06351, Korea; 5Stem Cell & Regenerative Medicine Institute, Samsung Medical Center, Seoul 06351, Korea; 6Seoul Jun Research Center, Seoul Jun Rehabilitation Clinic, Seoul 06737, Korea

**Keywords:** osteogenic differentiation, adipogenic differentiation, Runx2, PPARγ, 18α-glycyrrhetinic acid, T0070907

## Abstract

The inhibition of the aberrant differentiation of tendon-derived stem cells (TDSCs) is a major target for the regeneration of damaged tendon tissues, as tendinopathy can be caused by the aberrant differentiation of TDSCs. We investigated whether the possible aberrant differentiation of TDSCs can be prevented by using adequate inhibitors. TDSCs extracted from chemically induced tendinopathy and injury-with-overuse tendinopathy models were cultured with 18α-glycyrrhetinic acid (AGA) and T0070907 to block osteogenic differentiation and adipogenic differentiation, respectively. The optimal dose of AGA decreased the osteogenic-specific marker Runx2 (Runt-related transcription factor 2), and T0070907 blocked the adipogenic-specific marker peroxisome proliferator-activated receptor gamma (PPARγ) in mRNA levels. We also found that AGA induced tenogenic differentiation in mRNA levels. However, T0070907 did not affect the tenogenic differentiation and regenerative capacity of TDSCs. We expect that optimal doses of AGA and T0070907 can prevent tendinopathy by inhibiting osteogenic and adipogenic differentiation, respectively. In addition, AGA and T0070907 may play important roles in the treatment of tendinopathy.

## 1. Introduction

Degenerative tendinopathy is a pathologic change in tendon tissues caused by repetitive traumatic injury and characterized by the formation of non-tendinous tissues in the tendon matrix [1]. From histopathology in degenerative tendinopathy, calcifications, fibrocartilaginous or osseous metaplasia, and fatty degeneration can be found in addition to the deposition of mucoid material [2]. These abnormal non-tendinous tissues in the tendon contribute to widespread inferior biomechanics in the tendon [3], rendering the patients with tendinopathy physically disabled.

After the identification of the tendon-derived stem cells (TDSCs) and the unveiling of their characteristics, TDSCs have been thought to be concerned in the development and repair of degenerative tendinopathy [4]. The aberrant chondrogenic or adipogenic differentiation of TDSCs is related to the development of tendinopathy [4,5]. Chondrogenic differentiation was induced by kartogenin application [4], adipogenic differentiation after tendon injury, and osteogenic differentiation by chemical stress [5].

The major transcription factors that have key roles in the differentiation of mesenchymal stem cells into osteocytes are Runx2 and Osterix [6]. Runx2 is a transcription factor that is essential for osteoblast differentiation and chondrocyte maturation, and the inhibition of Runx2 reduces bone formation [7,8,9,10,11]. Peroxisome proliferator-activated receptor gamma (PPARγ) is an adipogenic regulator, which plays a role in lipid metabolism [12,13,14]. The inhibition of PPARγ by drugs or biologic molecules inhibits adipocyte differentiation [15,16,17]. Several studies have demonstrated that the suppression of PPARγ regulates adipogenic differentiation in mesenchymal stem cells by Wnt signals [18,19,20].

Several biological agents and stem cells including platelet-rich plasma and adipogenic stem cells have been evaluated to reverse or prevent degenerative tendinopathy. Several studies reported that stem cells improved clinical outcomes in the patients with tendinopathy. Lee et al. demonstrated injection of allogenic adipose-derived stem cells improved pain for patients with lateral epicondylopathy [21], and Usuelli et al. reported that intratendinous adipose-derived stromal vascular fraction or platelet rich plasma injection were safe, effective treatments for recalcitrant Achilles tendinopathy [22]. Kim et al. reported platelet rich plasma improved rotator cuff tendinopathy according to its components [23]. However, the mechanism of therapeutic efficacy was not evident. The biologic agents are focused on enhancing the regeneration of damaged tendon tissues by the differentiation of TDSCs to tenocytes and the secretion of collagens from tenocytes. The inhibition of the aberrant differentiation of TDSCs will also be a major target for the regeneration of damaged tendon tissues, because the induction of tendinopathy can be achieved by the aberrant differentiation of TDSCs due to several factors [5]. We hypothesized that tendinopathy can be reversed or prevented by inhibition of this aberrant differentiation by the stimulation of adequate factors. We investigated whether the possible aberrant differentiation processes of TDSCs, osteogenic and adipogenic differentiation, can be reversed by blocking these pathways.

## 2. Results

### 2.1. Effects of AGA and T0070907 on Cytotoxicity Of TDSCs

In a search for the optimal doses of inhibitors for Runx2 and PPARγ, TDSCs were cultured with inhibitors of various concentrations (0, 1, 10, 25, 50, 100 μM) for 5 days. We found that over 50 μM of 18α-glycyrrhetinic acid (AGA) remarkably decreased the surviving percentage of cells in the AGA treatment group compared to those in the control group. Therefore, we used 25 μM of AGA as the cytotoxic dose killing under 50% of TDSCs extracted from the chemically induced tendinopathy model (Figure 1a,b).

In addition, we found that over 25 μM of T0070907 decreased the surviving percentage of cells in the T0070907 treatment group compared to those in the control group. Thus, we used 10 μM of T0070907 as the cytotoxic dose killing under 50% of TDSCs extracted from the injury-with-overuse tendinopathy model (Figure 1c,d).

### 2.2. AGA Blocked Osteogenic Differentiation and Inhibited Expression of Runx2 at the mRNA Level

From our previous study, TDSCs extracted from the chemically induced tendinopathy model were more likely to undergo osteogenic differentiation compared to normal TDSCs, and TDSCs extracted from the injury-with-overuse tendinopathy model were more likely to undergo adipogenic differentiation compared to normal TDSCs [5].

Thus, we investigated whether AGA could block osteogenic differentiation. For osteogenic differentiation evaluation, AGA was added to TDSCs extracted from chemically induced tendinopathy models, whereas controls were cultured with DMSO. Cells were stained with Alizarin Red S and photographed under a light microscope.

When the results of four lots were combined and confirmed, relative calcium deposits in the AGA group were significantly decreased compared to the control group (*p* < 0.001). Although calcium deposits were markedly observed in 5T-1 and 5T-4 cells compared to 5T-2 and 5T-3 cells, AGA consistently inhibited osteogenic induction in all cells (5T-1, 2, 4, *p* < 0.001; 5T-3, *p* < 0.01) (Figure 2A,B).

To investigate the effect of AGA, we performed qRT-PCR to analyze the mRNA levels of Runx2, which is a marker for osteogenic differentiation. AGA consistently decreased the mRNA levels of Runx2 throughout all of the cells (*p* < 0.001). Although decreasing level of each lot differed, mRNA levels of Runx2 were decreased in the AGA group compared with those in the control group (5T-1, 5T-2, *p* < 0.05; 5T-3, *p* < 0.01) (Figure 2c). However, Western blot analysis showed no significant differences in the protein levels of Runx2 between AGA and the control group (Figure 2D). This result suggests that AGA blocks osteogenic differentiation by inhibiting the mRNA expression of Runx2 in TDSCs.

### 2.3. AGA Induced Tenogenic Differentiation and Regenerative Capacity of TDSCS in Part

Furthermore, we evaluated the expression of tenogenic markers and confirmed the effect of AGA on the tenogenic differentiation and regenerative capacity of TDSCs. Tenomodulin, scleraxis, and tenascin C were used as tenogenic differentiation markers, and collagen type I and collagen type III were used to confirmed regenerative capacity of TDSCs. AGA increased the mRNA level of tenomodulin and scleraxis in the 5T-2, 3, 4 cells (tenomodulin, 5T-2, 3, 4, *p* < 0.001; scleraxis, 5T-2, 3, 4, *p* < 0.001), whereas it slightly decreased those in the 5T-1 cells (tenomodulin, 5T-1, *p* < 0.001; scleraxis, 5T-1, *p* < 0.01). After the AGA treatment, the mRNA levels of tenascin C in the AGA group were significantly increased compared to those in the control group (5T-1, 3, *p* < 0.05; 5T-2, *p* < 0.01). Unexpectedly, AGA could not increase the mRNA levels of collagen type I, except in the 5T-4 cells. AGA increased the mRNA level of collagen type I in the 5T-4 cells, whereas it decreased those in the 5T-1, 2, and 3 cells. Except for some increase in 5T-3 (*p* < 0.001), the mRNA levels of collagen type III in the AGA group did not differ from the control group (Figure 3A). Tenogenic differentiation markers showed the increasing pattern in the mRNA levels after AGA treatment (tenomodulin, scleraxis, *p* < 0.05; tenascin C, *p* < 0.001), whereas the AGA treatment had no effects on regenerative capacity (Figure 3B). Similarly, Western blot analysis showed that AGA increased protein levels of collagen type I in 5T-4 cells (Figure 3C). Our study showed that the mRNA and protein levels of Runx2 were lower in the 5T-4 cells than those of any other cells (mRNA level, *p* < 0.05; protein level, *p* < 0.001) (Figure 2C,D), and that AGA could induce tenogenic regeneration only in those cells. Otherwise, Western blot analysis showed no significant differences in the protein levels of collagen type I of the 5T-1, 2, and 3 cells between the AGA and control groups. In addition, any significant differences were not observed in the protein levels of collagen type III and tenomodulin throughout all of the cells (Figure 3C).

### 2.4. T0070907 Blocked Adipogenic Differentiation and Inhibited Expression of PPARγ at the mRNA Level

We also investigated whether T0070907 could block adipogenic differentiation. For adipogenic differentiation, T0070907 was added to TDSCs extracted from injury-with-overuse tendinopathy models, whereas controls were cultured in the adipogenic medium with DMSO. Cells were stained with Oil Red O and photographed under a light microscope.

The expression of lipid droplets stained with Oil Red O in the control group was markedly increased, whereas lipid droplets were decreased in the T0070907 group (*p* < 0.001). Although lipid droplets were observed more in 15T-4 compared to other cells, T0070907 consistently inhibited osteogenic induction in all cells (15T-1, 2, 4, *p* < 0.001; 15T-3, *p* < 0.01) (Figure 4A,B).

Next, qRT-PCR was performed to investigate the effect of T0070907 by analyzing the mRNA levels of PPARγ, which is a marker for adipogenic differentiation. T0070907 decreased the mRNA levels of PPARγ in all of the cells (*p* < 0.001). Although there was a difference in decreasing level of each lot, the PPARγ level in the T0070907 group was significantly lower than that in the control group (15T-1, 3, *p* < 0.001; 15T-2, 4, *p* < 0.05) (Figure 4C). However, Western blot analysis showed no significant differences in the protein levels of PPARγ between T0070907 and the control group (Figure 4D). This result suggests that T0070907 has an inhibitory effect on adipogenic differentiation by inhibiting the mRNA expression of PPARγ.

### 2.5. T0070907 Had No Effect on the Tenogenic Differentiation and Regenerative Capacity of TDSCs

We also examined the expression of tenogenic markers and confirmed the effects of T0070907 on the tenogenic differentiation and regenerative capacity of TDSCs. Unexpectedly, only 15T-3 cells showed increased mRNA levels of tenascin C in the presence of T0070907, and the other cells were decreased by T0070907 in terms of the mRNA levels of tenascin C (15T-1, *p* < 0.01; 15T-2, 15T-3, *p* < 0.001; 15T-4, *p* < 0.05). After T0070907 treatment, the mRNA levels of collagen type I, tenomodulin and scleraxis were increased only in 15T-4 cells (collagen type I, scleraxis, *p* < 0.01; tenomodulin, *p* < 0.001), whereas the mRNA levels of collagen type I, tenomodulin, and scleraxis were decreased in 15T-1, 2 (tenomodulin, 15T-1, *p* < 0.001; 15T-2, *p* < 0.05; scleraxis, 15T-1, *p* < 0.01; 15T-2, *p* < 0.001). T0090907 decreased the mRNA levels of collagen type III in 15T-1, 2 cells (15T-1, *p* < 0.05; 15T-2, *p* < 0.01), but did not change the mRNA levels of collagen type III in 15T-3, 4 cells (Figure 5A). There was no significant change in the mRNA level of tenogenic markers after T0070907 treatment, even the mRNA level of collagen type III and scleraxis was slightly decreased (collagen type III, scleraxis, *p* < 0.05) (Figure 5B). The protein levels of collagen type I also were increased only in 15T-4 cells (*p* < 0.001). Otherwise, Western blot analysis showed no significant differences in the protein levels of collagen type III and tenomodulin between the T0070907 and control groups (Figure 5C).

## 3. Discussion

A previous study found evidence that TDSCs extracted from the chemically induced tendinopathy model were more likely to undergo osteogenic differentiation compared to normal TDSCs, and TDSCs extracted from the injury-with-overuse tendinopathy model were more likely to undergo adipogenic differentiation compared to normal TDSCs [5]. Building upon these findings, this study sought to investigate the therapeutic potential of suppressing the aberrant differentiation processes in TDSCs from tendinopathy through the inhibitors AGA and T0070907.

We identified a Runx2 inhibitor, AGA, which blocks the osteogenic differentiation of TDSCs from chemically induced tendinopathy, and a PPARγ antagonist, T0070907, that inhibits the adipogenic differentiation of TDSCs from injury-with-overuse tendinopathy. This indicates that AGA and T0070907 can prevent tendinopathy by inhibiting this aberrant differentiation. However, we found that AGA induced tenogenic differentiation in mRNA levels, but not in protein levels. T0070907 did not affect tenogenic differentiation in either mRNA or protein levels. In addition, the regenerative capacity of TDSCs was increased by AGA only in 5T-4, whereas there was no effect of T0070907 on the regenerative capacity of TDSCs in any other cells.

AGA was used in several studies as an inhibitor of gap junctional communication or osteoblastic differentiation. Guo et al. [24] showed that AGA blocked gap junctional communication mediated by connexin 43 in lung cell, and Talbot et al. [25] suggested that treatment with AGA inhibits the differentiation of human bone marrow stromal cells into osteoblasts. In addition, Jeong et al. [26] showed that AGA has an inhibitory effect on osteoblastic differentiation and inhibits the transcriptional activity of Runx2. The role of T0070907 in suppressing PPARγ activity has been demonstrated in many studies [27,28,29,30]. It has also been shown that T0070907 treatment has significant antitumor effects in various cancer cell types, and these effects are explained by the PPARγ pathway [31,32,33]. Although AGA and T0070907 are strong inhibitors of aberrant differentiation in a wide range of cells, the inhibition of aberrant differentiation in TDSCs by AGA and T0070907 remains unclear. Thus, we suggested that AGA and T0070907 inhibit the osteogenic and adipogenic differentiation of TDSCs, respectively.

We found the optimal dose of each inhibitor—25 μM of AGA for Runx2 and 10 μM of T0070907 for PPARγ. Previous studies also used the same doses of inhibitors to observe aberrant differentiation [26,30,34,35], which was consistent with our results. Over 25 μM of AGA killed more than 50% of cells due to the cytotoxicity of this inhibitor. In addition, the dose of T0070907 to kill over 50% of TDSCs is greater than 10 μM. For this reason, we adopted the maximal dose of each inhibitor that does not show severe cytotoxicity to confirm the inhibitory effects of these inhibitors.

Optimal doses of AGA and T0070907 inhibited the osteogenic differentiation of TDSCs from chemically induced tendinopathy and the adipogenic differentiation of TDSCs from injury-with-overuse tendinopathy, respectively. Therefore, we expect that optimal amounts of AGA and T0070907 can prevent tendinopathy by inhibiting this aberrant differentiation. In addition, we expect that these antagonists can play an important role in the treatment of tendinopathy. We confirmed the expression level of the tenogenic markers after treatment of AGA and T0070907. To account for key tendon-related components, this study utilized a range of tenogenic markers including tenomodulin, scleraxis, tenascin C, collagen type I, and collagen type III [36]. Collagen type I, collagen type III, and tenascin C are components of the extracellular matrix of tendon, and tenomodulin and scleraxis are markers known to be involved in the tenogenic differentiation [37]. mRNA level of collagen type I and collagen type III increase in pathologic human tendon and might be associated with tendon regeneration [38].

Although both AGA and T0070907 had effects in the inhibition of the aberrant differentiation of TDSCs, only AGA induced tenogenic differentiation of TDSCs in mRNA levels. We also confirmed that AGA induced tenogenic regeneration only in 5T-4, whose basal Runx2 expression was relatively low in both mRNA and protein level. Therefore, we think that AGA might increase tenogenic regenerative capacity but have no effect on cells in which Runx2 was highly expressed. Otherwise, T0070907 did not induce both tenogenic regeneration and differentiation. T0070907 had no effect on tenogenic regeneration and differentiation in protein level, but might also play some role in inhibiting tenogenic differentiation and regeneration in mRNA level, as well as osteogenic and adipogenic differentiation. We think that this phenomenon might be due to the cytotoxicity of this inhibitor. Furthermore, both AGA and T0070907 have a tendency to inhibit cell proliferation capacity because of their cytotoxicity [31,34]. For the same mechanism, in the case of TDSCs from tendinopathy, AGA and T0070907 might inhibit the proliferative capacity as well. Liu et al. [39] reported that the correlation of mRNA expression and protein expression was often inconsistent. Contrary to expectations, change of protein level could not have occurred immediately and delayed after mRNA level was changed. This was because it takes some time to maturate, export, and translate mRNA. Post-transcriptional regulation is important for protein level because regulation of transcript level would be slow. Or, it may be the result of the longer half-life of the protein than the RNA. It may have been difficult to observe changes in protein expression because the proteins that had already accumulated before the RNA levels decreased still remain in the cells. Post-translational modifications, such as phosphorylation, acetylation, glycosylation, and ubiquitination may have an effect on the stability of the protein. Specifically, Zaytseva et al. [31] showed that T0070907 did not have an effect on protein levels of PPARγ, which is consistent with our results. They also showed that T0070907 altered the phosphorylation of PPARγ, which suggests that T0070907 might affect the activity of upstream levels. Our study should further evaluate the phosphorylation of PPARγ to follow up on this previous research. Thus, further study is required to find different inhibitors or to perform the experiments with RNAi that are less cytotoxic and induce tenogenic differentiation and regeneration of TDSCs.

From our study, we showed that tendinopathy can be reversed or prevented by inhibition of aberrant differentiation by the stimulation of adequate factors. This was the first study to suggest the possibility that inhibition of the aberrant differentiation of TDSCs could be a therapeutic tool. We hope that our study will be a cornerstone to develop a new regenerative therapeutic agent for tendinopathy.

## 4. Materials and Methods

### 4.1. Extraction of TDSCs from Tendinopathy Models

Eight 4-week-old Sprague-Dawley rats were obtained from Orient Bio Inc. (Gapyeong, Korea) and housed in the Samsung Biomedical Research Institute for 1 week. After 1 week, four rats were given a single 900 mg/kg dose of ofloxacin (Ofloxacin, Myungmoon Pharm. Co. Ltd., Seoul, Korea) to create a chemically induced tendinopathy model, and four rats underwent partial Achilles tendon injury at 9 weeks of age and treadmill training for 5 weeks to create an injury-with-overuse tendinopathy model. The detailed protocols for the chemically induced and injury-with-overuse tendinopathy models were described in previous studies [5,40,41].

Preparation of tendinopathy models was confirmed by paraffin block and hematoxylin and eosin staining of tendon tissues extracted from the right side Achilles tendons, which showed an irregular fibrillary pattern of collagen fibers and pyknotic nuclei in the tenocytes.

A primary culture of TDSCs was acquired by adding collagenase type I (Sigma-Aldrich, cat. C0130, St. Louis, MO, USA) and dispase (Sigma-Aldrich, cat. D4693) to the tendon tissues extracted from the left side Achilles tendon within 30 min, digesting the tissues for 2 h, and culturing the supernatants by centrifugation. The process of the primary culture of TDSCs was fully described in a previous study [5].

TDSCs were identified by the immunofluorescent staining of tenomodulin, a cell marker of tendon-derived cells, and OCT4 (octamer-binding transcription factor 4), SSEA4 (stage-specific embryonic antigen-4), and nucleostemin, which are stem cell markers. Positive staining of tenomodulin represented teno-lineage cells and positive staining of OCT4, SSEA4, and nucleostemin represented characteristics of stem cells.

Five-week-old rats were used to make the chemically induced tendinopathy model and 15-week-old rats were used to make the injury-with-overuse tendinopathy model. For each condition, four 5-week-old and four 15-week-old rat models were used to extract cells for investigation of osteogenic and adipogenic differentiation, respectively. The cells of each of the four rats under the condition of 5-week-old, chemically induced tendinopathy were labelled as 5T-1, 5T-2, 5T-3, and 5T-4. Similarly, the cells of each of the four rats under the condition of 15-week-old, injury-with-use tendinopathy were labelled as 15T-1, 15T-2, 15T-3, and 15T-4.

### 4.2. Selection of Optimal Doses for Blocking Agents

We selected T0070907 (Sigma-Aldrich) as a PPARγ inhibitor for blocking adipogenic differentiation and 18α-glycyrrhetinic acid (AGA, Sigma-Aldrich) as a Runx2 inhibitor for blocking osteogenic differentiation. T0070907 was selected on the basis of a previous article showing the adipogenic blocking capacity [27], and AGA was selected on the basis of a previous study showing the osteogenic blocking capacity [26].

For the selection of optimal doses for these blocking agents, a cytotoxicity assay was performed. The cytotoxicity assay was performed using Cell Counting Kit (CCK)-8 solution (Dojindo Molecular Technologies Inc., Gaithersburg, MD, USA) according to the manufacturer’s protocol. T0070907 and AGA were dissolved with 100% dimethyl sulfoxide (DMSO, Sigma Aldrich) to make a stock solution and then diluted with Dulbecco’s modified Eagle’s medium (DMEM) to a final DMSO concentration of 1%. After seeding 2 × 10^3^ TDSCs with 100 μL DMEM and 10% fetal bovine serum (FBS) on 96-well culture plates, 0 μM, 1 μM, 10 μM, 25 μM, 50 μM, or 100 μM of T0070907 or AGA were added to each well 24 h later. The culture plates were incubated for 5 days and were then incubated for another 3 hours after adding 10 μL of CCK-8 solution to each well. The optical densities of the wells were measured at 450 nm using a microplate reader and normalized by DMSO controls. This cytotoxicity assay was triplicated for consistency.

### 4.3. Inhibition of Aberrant Differentiation by Blocking Agents

To find the aberrant osteogenic differentiation and its inhibition by blocking agents, TDSCs extracted from chemically induced tendinopathy models were cultured on 24-well culture plates until the cells were confluent under standard growth medium (DMEM with 10% FBS). Then, cells were cultured on the osteogenic medium (Thermo Fisher Scientific, cat. A10072-01, Waltham, MA, USA) for 21 days. The control group was treated with DMSO only, and the treatment group was exposed to 25 μM of AGA with DMSO concentrations for 21 days.

The media were removed from the cell culture plates, and the cells were washed with PBS, fixed in 4% paraformaldehyde for 30 min at room temperature, and then washed with phosphate-buffered saline (PBS) three times. Next, the cells were incubated with 2% Alizarin Red S staining solution (Sciencell Research Laboratories, cat. 0223, Carlsbad, CA, USA) for 30 min at room temperature. Samples were washed with distilled water three times, and 1 mL of distilled water was added to prevent the cells from drying out. Stained samples were observed using a light microscope. Stained calcium deposits were shown in red, implying osteogenic differentiation.

To find the adipogenic differentiation and its inhibition by blocking agents, TDSCs extracted from injury-with-overuse tendinopathy were cultured using the same methods but on the adipogenic media (Thermo Fisher Scientific, cat. A10070-01) instead of osteogenic media. The cells were cultured on 24-well culture plates. The control group was treated with DMSO alone, and the treatment group was exposed to 10 μM of T0070907 with DMSO concentrations for 21 days.

The media were removed from the cell culture plates, and the cells were washed with PBS, fixed in 4% paraformaldehyde for 30 min at room temperature, and then washed with PBS three times. Next, the cells were incubated with 0.3% Oil Red O solution (Sigma Aldrich, cat. O1391) for 50 min at room temperature. Samples were washed with distilled water three times, and 1 mL of distilled water was added to prevent the cells from drying out. Stained samples were observed using a light microscope. Stained lipid droplets were shown in red, implying adipogenic differentiation. To increase the reliability of the results, all experiments were repeated four times.

The extent of lipid droplets and calcium deposits in microcopy images were measured using ImageJ, and five fields were analyzed for each group.

### 4.4. Quantitative Real-Time Polymerase Chain Reaction (qRT-PCR)

TDSCs (5 × 10^5^ cells) were seeded and cultured in 75T flasks for 7 days under the same conditions as described for the cytotoxicity assay. Total RNA was isolated using an AccuPrep Universal RNA Extraction kit (Bioneer, cat. K-3141, Daejeon, Korea) according to the manufacturer’s protocol. Then, cDNA (complementary DNA) was obtained using 4 μg of RNA with the addition of superscript II reverse transcriptase (Invitrogen, cat. 18064-014, Waltham, MA, USA) to 40 μL of reaction buffer for 2 h at 42 °C. Real-time polymerase chain reaction (RT-PCR) was performed with a Quantstudio 6 Flex Real-Time PCR System (Thermo Fisher Scientific) with a total volume of 25 μL including specific primers, and we used a QuantiTect SYBR Green PCR Kit (Qiagen, cat. 204143, Hilden, Germany) with 30 ng of cDNA. PCR conditions were as follows: 95 °C for 15 min, 40 cycles at 95 °C for 15 s, optimal annealing temperature for 30 s, and 72 °C for 30 s. The following primer sequences were used (Genotech Corp., Daejeon, Korea): Col1A1 (forward 5’-GGCCCAGAAGAACTGGTACA-3’, reverse 5’-GGCTGTTCTTGCAGTGGTAG-3’); Col3A1 (forward 5’-GATGGCTGCACTAAAC-3’, reverse 5’-CGAGATTAAAGCAAGAG-3’); Scleraxis (forward 5’-AACACGGCCTTCACTGCGCTG-3’, reverse 5’-CAGTAGCACGTTGCCCAGGTG-3’); tenomodulin (forward 5’-CCATGCTGGATGAGAGAGGT-3’, reverse 5’-CTCGTCCTCCTTGGTAGCAG-3’); tenascin C (forward 5’-GACCTGGCCTATGAGTAC-3’, reverse 5’-AGCACGGGTGTTTTATAGC-3’); Runx2 (forward 5’-TGATGACACTGCCACCTCTGACTT-3’, reverse 5’-TGGATA GTGCATTCGTGGGTTGGA-3’); PPARγ (forward 5’-CGAGCCCTGGCAAAGCATTTGT AT-3’, reverse 5’-TGTCTTTCCTGTCAAGATCGCCCT-3’); β-actin (forward 5’-TTGCTGACAGGATGCAGAAGGAGA-3’, reverse 5’-ACTCCTGCTTGCTGATCCACATCT-3’) [37,42]. Each gene was standardized to β-actin, and relative expression levels were measured using the 2^−ΔΔCt^ method. Error bars represent the standard deviation of mean. For statistical analysis, SPSS statistics 23 software (IBM Corp., Armonk, NY, USA) was used, and ANOVA analysis was conducted to calculate significance. *p* < 0.05 was considered statistically significant.

### 4.5. Western Blot

Western blots were performed to quantify the expression levels of Runx2, PPARγ, collagen type I, collagen type III, and tenomodulin proteins. TDSCs were seeded at the density of 5 × 10^5^ cells in 75T flasks and cultured for 7 days with the same conditions as indicated for the cytotoxicity assay. Cell lysis was performed with RIPA cell lysis buffer (BIOSESANG, cat. R2002, Sungnam, Gyeonggi, Korea) including a protease inhibitor cocktail (Amresco, cat. M250-1 mL, Solon, OH, USA), and the supernatants were obtained by centrifugation at 14,000× *g* for 10 min. Protein concentrates collected from the supernatants were quantified by Bradford assay (Sigma Aldrich, cat. B6916), and then 40 μg of the concentrates were loaded into 4–12% Bis-Tris gels (Life Technologies, cat. NW04120, Carlsbad, CA, USA). The loaded samples were transferred to nitrocellulose membranes (Life Technologies, cat. IB23001) and blocked with 5% bovine serum albumin for 1 h at room temperature. Membranes were incubated with primary antibodies (Runx2, 1:200, Cell Signaling Technology, cat. 8486, MA, USA; PPARγ, 1:200, Abcam, cat. ab41928, Cambridge, UK; collagen type I antibody, 1:500, Abcam, cat. ab292; collagen type III antibody, 1:100, Thermo Fisher Scientific, cat. MA1-22147; tenomodulin antibody, 1:200, Santa Cruz Biotechnology, cat. SC-98875, CA, USA; β-actin, 1:2000, Santa Cruz Biotechnology, cat. SC-81178) overnight at 4 °C. Next, they were washed with 0.1% Tris-buffered saline three times for 10 min, and then incubated with secondary antibody (goat anti-mouse, 1:2000, Santa Cruz Biotechnology, cat. SC-2005; goat anti-rabbit, 1:1000, Santa Cruz Biotechnology, cat. SC-2004) for 1 hour at room temperature. After washing with 0.1% Tris-buffered saline three times for 10 min, Western bands were obtained with a gel imaging system (Amersham Imager 600, GE Healthcare, Buckinghamshire, UK), and their intensities were measured with ImageJ software and then normalized to β-actin.

All of the protocols were approved by our institutional IACUC (Institutional Animal Care and Use Committee) (approval no. 20170404001).

## 5. Conclusions

This study is the first piece of research that shows the inhibition of aberrant differentiation in TDSCs using AGA and T0070907. These results suggest that AGA and T0070907 can be a potential treatment for tendinopathy. However, further studies with animal experiments are necessary to develop a new treatment based on our results. These inhibitory effects of AGA and T0070907 provide insight into the development of treatments for tendinopathy.

In addition, TDSCs have the potential to show another aberrant differentiation such as chondrogenic differentiation as well as osteogenic or adipogenic differentiation [4,43,44,45]. Future studies will need to investigate how the chondrogenic differentiation of TDSCs is related to the development of tendinopathy. These results suggest that inhibition of the aberrant differentiation could be prevent tendinopathy and a therapeutic tool.

## Figures and Tables

**Figure 1 ijms-21-02687-f001:**
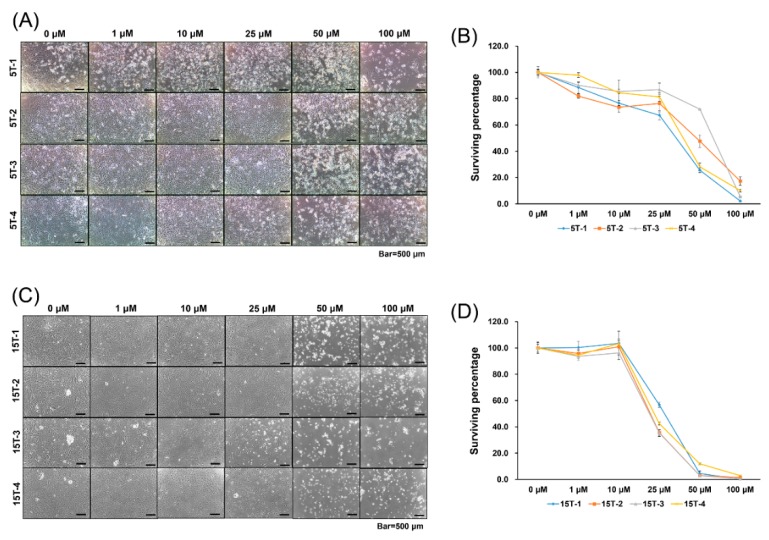
Cytotoxicity assay of 18α-glycyrrhetinic acid (AGA) and T0070907. Representative microscopic images of tendon-derived stem cells (TDSCs) cultured with various concentrations of (**A**) AGA and (**C**) T0070907. 5T-1, 5T-2, 5T-3, and 5T-4 stem cells were collected from different rat specimens as 5-week-old, chemically induced tendinopathy donors. In the same manner, 15T-1, 15T-2, 15T-3, and 15T-4 cells were collected from different rat specimens as 15-week-old, injury-with-overuse tendinopathy donors. TDSCs were cultured with various concentrations (0, 1, 10, 25, 50, 100 μM) for 5 days. Cytotoxicity of (**B**) AGA and (**D**) T0070907 was measured by Cell Counting Kit (CCK)-8 assay. The optimal dose of AGA and T0070907 killing less than 50% of TDSCs was 25 μM and 10 μM, respectively. In (**B**,**D**), error bars represent the standard deviation of mean.

**Figure 2 ijms-21-02687-f002:**
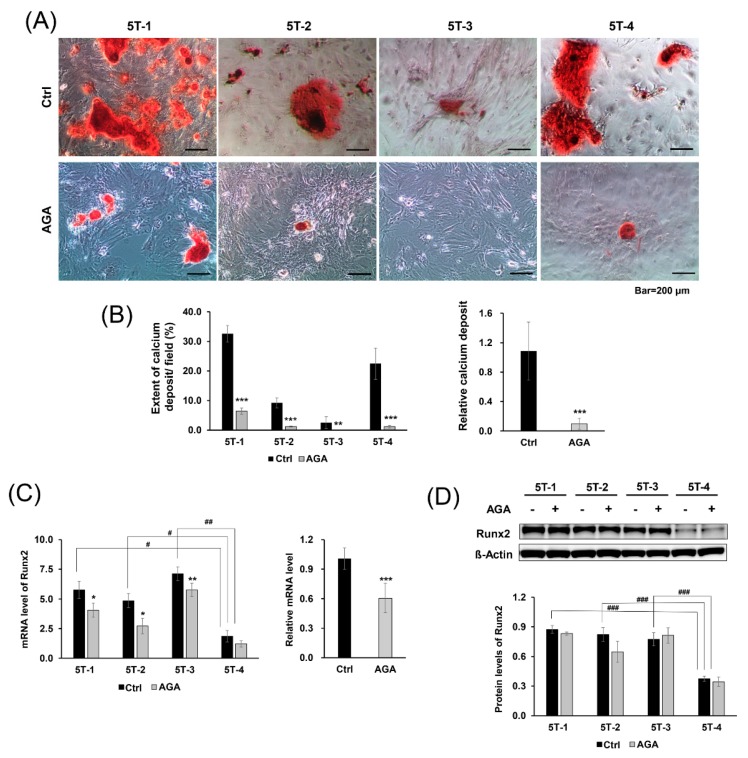
Effect of AGA on osteogenesis. (**A**) Representative microscopic images of TDSCs cultured with and without AGA for 21 days. TDSCs were incubated with 2% Alizarin Red S staining solution and (**B**) extent of calcium deposit per field measured stained areas using ImageJ. The relative calcium deposit represents the value divided the extent of calcium deposit in the AGA treatment group by the extent of calcium deposit in the control group in all of the cells. Calcium deposits were decreased in the presence of AGA. AGA effects on the expression of Runx2 were measured by (**C**) qRT-PCR and (**D**) Western blot. (**C**) qRT-PCR analysis showed decreased mRNA levels of Runx2 in the presence of AGA, (**D**) whereas Western blot analysis presented no significant change in protein levels of Runx2. The relative mRNA level represents the value divided Runx2 mRNA level of the AGA treatment group by Runx2 mRNA level of the control group in all of the cells. mRNA level of Runx2 represents mRNA expression levels standardized to β-actin, and protein level of Runx2 represents normalized protein expression level by β-actin. Error bars represent the standard deviation of mean. * *p* < 0.05, ** *p* < 0.01, *** *p* < 0.001, compared with AGA group and control group. # *p* < 0.05, ## *p* < 0.01, ### *p* < 0.001, compared with 5T-4 cell and other cells.

**Figure 3 ijms-21-02687-f003:**
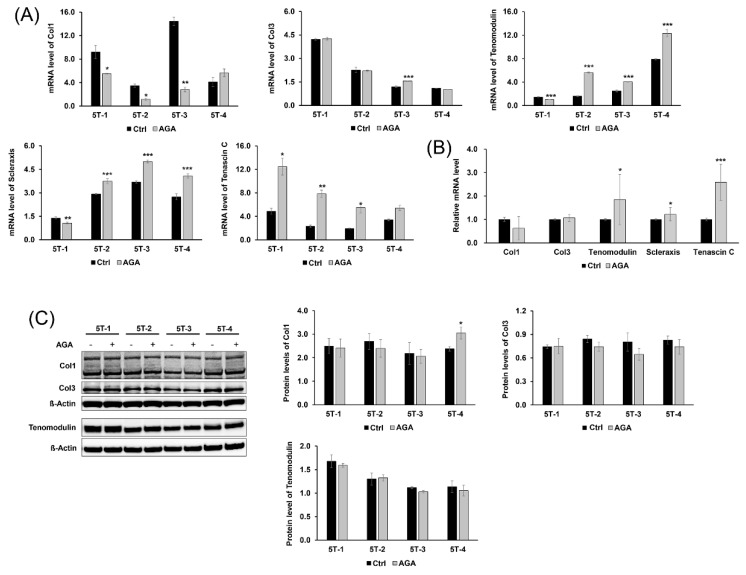
Effect of AGA on tenogenic differentiation and regenerative capacity of TDSCs. (**A**) mRNA levels of collagen type I (Col1), collagen type III (Col3), tenomodulin, scleraxis, and tenascin-C were measured by qRT-PCR. (**B**) AGA increased mRNA levels of tenogenic differentiation marker in the AGA group, whereas the treatment had no effects on regenerative capacity. The relative mRNA level represents the value divided mRNA level of the AGA treatment group by mRNA level of the control group in all of the cells. (**C**) Western blot analysis was used to measure protein levels of collagen type I, collagen type III, and tenomodulin. AGA had no effects on tenogenic differentiation and regenerative capacity in the expression of protein levels. mRNA level represents mRNA expression levels standardized to β-actin and protein level represents normalized protein expression level by β-actin. Error bars represent the standard deviation of mean. * *p* < 0.05, ** *p* < 0.01, *** *p* < 0.001, compared with AGA group and control group.

**Figure 4 ijms-21-02687-f004:**
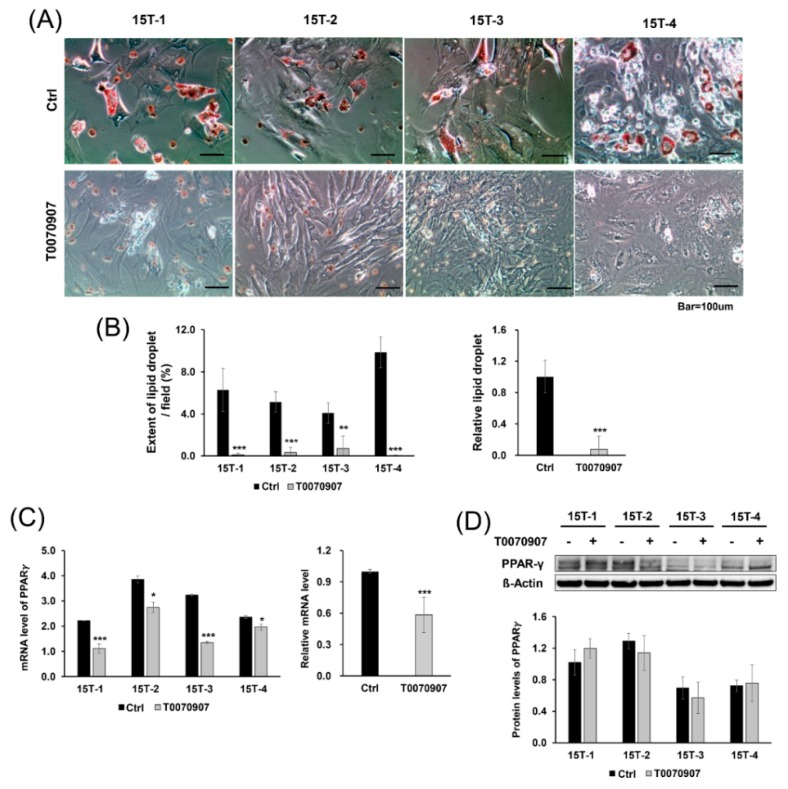
Effect of T0070907 on adipogenesis. (**A**) Representative microscopic images of TDSCs cultured with and without T0070907 for 21 days. TDSCs were incubated with 0.3% Oil Red O solution, and (**B**) extent of lipid droplet per field was measured the stained areas using ImageJ. The relative lipid droplet represents the value divided the extent of lipid droplet in the T0070907 treatment group by the extent of lipid droplet in the control group in all of the cells. Lipid droplets were decreased in the presence of T0070907. T0070907 effects on the expression of peroxisome proliferator-activated receptor gamma (PPARγ) were measured by (**C**) qRT-PCR and (**D**) Western blot. (**C**) qRT-PCR analysis showed decreased mRNA levels of PPARγ in the presence of T0070907, (**D**) whereas Western blot analysis presented no significant change in protein levels of PPARγ. The relative mRNA level represents the value divided PPARγ mRNA level of the T0070907 treatment group by PPARγ mRNA level of the control group in all of the cells. mRNA level of PPARγ represents mRNA expression levels standardized to β-actin, and protein level of PPARγ represents normalized protein expression level by β-actin. Error bars represent the standard deviation of mean. * *p* < 0.05, ** *p* < 0.01, *** *p* < 0.001, compared with T0070907 group and control group.

**Figure 5 ijms-21-02687-f005:**
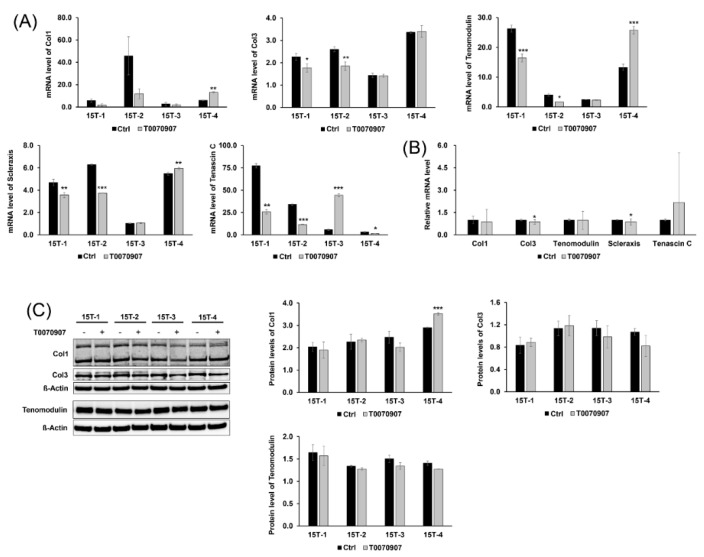
Effect of T0070907 on tenogenic differentiation and regenerative capacity of TDSCs. (**A**) mRNA levels of collagen type I, collagen type III, tenomodulin, scleraxis, and tenascin C were measured by qRT-PCR. (**B**) T0070907 were not increased mRNA levels of tenogenic markers. The relative mRNA level represents the value divided mRNA level of the T0070907 treatment group by mRNA level of the control group in the all of the cells. (**C**) Western blot analysis was used to measure protein levels of collagen type I, collagen type III, and tenomodulin. T0070907 had no effects on tenogenic differentiation and regenerative capacity in the expression of protein levels. mRNA level represents mRNA expression levels standardized to β-actin and protein level represents normalized protein expression level by β-actin. Error bars represent the standard deviation of mean. * *p* < 0.05, ** *p* < 0.01, *** *p* < 0.001, compared with T0070907 group and control group.

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
