# Peer review of "Recovery of Tendon Characteristics by Inhibition of Aberrant Differentiation of Tendon-Derived Stem Cells from Degenerative Tendinopathy"

_ijms, 2020, doi:10.3390/ijms21082687_

Round 1
Reviewer 1 Report
The authors have properly adressed the comments now.
Author Response
The authors have properly adressed the comments now.
We appreciate your comments.
Reviewer 2 Report
This is an interesting article in which the authors refer to the possibility of using two inhibitors to prevent aberrant differentiation of the tendons. This being one of the biggest medical problems in this type of injury. Allowing the reduction of surgical interventions and their complications. 1. In the article I miss more references, and more updated, many of them focused on the expression of Runx2 and PPARγ. 2. In some parts of the article the authors use the present tense, as in the beginning of section 2.3. 3. What is the reason for the fact that the authors have carried out the experiments for 5 days and have not carried out for more days to see the progression of the differentiation? 4. Authors should analyze microscopy images to provide absolute data, such as number and size of lipid droplets and calcium deposits. 5. The authors have carried out the statistical studies considering the cultures independently. In the case of primary cultures, the experiments should be analyzed as independent, but using an ANOVA test, so that the significance of the treatment would be clearer. 6. By grouping groups by treatment, the authors can relativize each point with the control and thus normalize the data, so that the clearest% inhibition can be observed. 7. In the charts the authors name the Control as Con, the best form is Ctrl. 8. The authors show that inhibitor T0070907 greatly reduces PPARγ mRNA expression, but not protein expression. How can the authors explain this data? Is it due to transcriptional modifications? It would be interesting for the authors to do a genetic study to determine if the inhibitors act at the genetic level. Or in your case explain your theory better. 9. The authors should develop a clearer discussion and conclusion of the data provided, taking into account the comments made in the previous pointsAuthor Response
This is an interesting article in which the authors refer to the possibility of using two inhibitors to prevent aberrant differentiation of the tendons. This being one of the biggest medical problems in this type of injury. Allowing the reduction of surgical interventions and their complications.
- In the article I miss more references, and more updated, many of them focused on the expression of Runx2 and PPARγ.
Line 51-58: We added more references focused on the expression of Runx2 and PPARγ. Reference has increased from seven to fifteen.
- In some parts of the article the authors use the present tense, as in the beginning of section 2.3.
Line 133-135: Tenomodulin, Scleraxis and Tenascin C were used as tenogenic differentiation markers, and collagen type I and collagen type III were used to confirmed regenerative capacity of TDSCs.
- What is the reason for the fact that the authors have carried out the experiments for 5 days and have not carried out for more days to see the progression of the differentiation?
In the previous paper [5], when the experiment was conducted to seed the same amount of cells and confirm the proliferation, we confirmed that cells were confluent in 5 days. We decided that it was difficult to tell whether the cells that die after 5 days were caused by drugs or overgrowth, so we didn’t carry out the CCK-8 assay for more than 5 days.
We first confirmed western blot after drug treatment for 5 days, but there was no change in protein level. So, we extended the period and confirmed qRT-PCR and western blot after drug treatment for 7 days. However, changes in RNA levels were observed, but no changed in protein levels were observed.
Line 373-374: TDSCs (5 × 105 cells) were seeded and cultured in 75T flasks for 7 days under the same conditions as described for the cytotoxicity assay.
Line 399-400: TDSCs were seeded at the density of 5 × 105 cells in 75T flasks and cultured for 7 days with the same conditions as indicated for the cytotoxicity assay.
We performed the drug treatment for 21 days to see the progression of the differentiation.
Line 347-350: Then, cells were cultured on the osteogenic medium (Thermo Fisher Scientific, cat. A10072-01, Waltham, MA, USA) for 21 days. The control group was treated with DMSO only, and the treatment group was exposed to 25 μM of AGA with DMSO concentrations for 21 days.
Line 361-362: The control group was treated with DMSO alone, and the treatment group was exposed to 10 μM of T0070907 with DMSO concentrations for 21 days.
- Authors should analyze microscopy images to provide absolute data, such as number and size of lipid droplets and calcium deposits.
Using Image J, we measured the extent of lipid droplets and calcium deposits in microcopy images and reflected them in the Figure. 2A and Figure. 4A.
Line 370-371: The extent of lipid droplets and calcium deposits in microcopy images were measured using Image J and five fields were analyzed for each group.
- The authors have carried out the statistical studies considering the cultures independently. In the case of primary cultures, the experiments should be analyzed as independent, but using an ANOVA test, so that the significance of the treatment would be clearer.
In accordance with the advice of reviewers, we proceeded all the analysis again with ANOVA test and applied the results to manuscript and all figures.
Line 394-396: For statistical analysis, SPSS statistics 23 software (IBM Corp., Armonk, NY, USA) was used and ANOVA analysis were conducted to calculate significance. p < 0.05 was considered statistically significant.
- By grouping groups by treatment, the authors can relativize each point with the control and thus normalize the data, so that the clearest% inhibition can be observed.
Figure 2A, 2B, 3A, 4A, 4B, and 5A: We compared the control group to the drug treatment group by combining the results of four lots. We reflected all the results in figures.
- In the charts the authors name the Control as Con, the best form is Ctrl.
In the charts, I renamed the Control as Ctrl instead of Con.
- The authors show that inhibitor T0070907 greatly reduces PPARγ mRNA expression, but not protein expression. How can the authors explain this data? Is it due to transcriptional modifications? It would be interesting for the authors to do a genetic study to determine if the inhibitors act at the genetic level. Or in your case explain your theory better.
The correlation of mRNA expression and protein expression is often inconsistent [39]. Contrary to expectations, change of protein level was not happened immediately and may be delayed after mRNA level was changed. Because it takes some times to maturate, export, and translate mRNA. Post transcriptional regulation are important for protein level because regulation of transcript level would be slow. Or it may be the result of the longer half-life of the protein than the RNA. It may have been difficult to observe changes in protein expression because the proteins that had already accumulated before the RNA levels decreased still remain in the cells. Post-translational modifications, such as phosphorylation, acetylation, glycosylation, ubiquitination may have effect on the stability of the protein.
We tried to get an antibody to check the phosphorylation, but we received a reply that it would take more than two months to import. Unfortunately, it is unlikely to be confirmed by the experiment.
- The authors should develop a clearer discussion and conclusion of the data provided, taking into account the comments made in the previous points
Line 279-288: Liu et al. [39] reported that the correlation of mRNA expression and protein expression is often inconsistent. Contrary to expectations, change of protein level was not happened immediately and may be delayed after mRNA level was changed. Because it takes some times to maturate, export, and translate mRNA. Post transcriptional regulation are important for protein level because regulation of transcript level would be slow. Or it may be the result of the longer half-life of the protein than the RNA. It may have been difficult to observe changes in protein expression because the proteins that had already accumulated before the RNA levels decreased still remain in the cells. Post-translational modifications, such as phosphorylation, acetylation, glycosylation, ubiquitination may have effect on the stability of the protein.
Line 294-298: From our study, we showed that tendinopathy can be reversed or prevented by inhibition of aberrant differentiation by the stimulation of adequate factors. This was the first study to suggest the possibility that inhibition of the aberrant differentiation of TDSCs could be a therapeutic tool. We hope that our study will be a cornerstone to develop a new regenerative therapeutic agent for tendinopathy.
Line 429-430: These results suggest that inhibition of the aberrant differentiation could be prevent tendinopathy and a therapeutic tool.
Reviewer 3 Report
In this study the authors, showed in a preliminary manner the effects of AGA and T0070907 to block osteblastic and adipogenic differentiation.
The project idea can be interest and the corrections done well improved the quality of the manuscript.
According to me, the manuscript can be considered acceptable for the publications with a minor revision.
In particular, to further improve the quality of the manuscript, the authors should discuss better the following points:
- In the introduction section, It 's better to increase the clinical evidences about the theraputic approaches used today for tendinopathy;
- in introduction section, the authors described the role of CBFA-1 as an osteblast markers. But, this is not reported in the figures or results. It's better to eliminate it from the introduction.
- In material and methods section please to insert the correct catalog number of Col1a-antibody used in WB.
- In the discussion section, I suggest to improve the final part highlight the important of the data obtained to improve the clinical practise.
It is better to check the English language and style. I suggest to check line 130.
Author Response
This study the authors, showed in a preliminary manner the effects of AGA and T0070907 to block osteblastic and adipogenic differentiation.
The project idea can be interest and the corrections done well improved the quality of the manuscript.
According to me, the manuscript can be considered acceptable for the publications with a minor revision.
In particular, to further improve the quality of the manuscript, the authors should discuss better the following points:
- In the introduction section, it's better to increase the clinical evidences about the theraputic approaches used today for tendinopathy;
- Line 60-66: Several studies reported that stem cells improved clinical outcomes in the patients with tendinopathy. Lee et al demonstrated injection of allogenic adipose-derived stem cells improved pain for patients with lateral epicondylopathy [21] and Usuelli et al reported that intratendinous adipose-derived stromal vascular fraction or platelet rich plasma injection were safe, effective treatments for recalcitrant Achilles tendinopathy [22]. Kim et al reported platelet rich plasma improved rotator cuff tendinopathy according to its components [23]. However, the mechanism of therapeutic efficacy was not evident.
- in introduction section, the authors described the role of CBFA-1 as an osteblast markers. But, this is not reported in the figures or results. It's better to eliminate it from the introduction.
- Line 52: We deleted the contents of CBFA-1 following the reviewer’s advice.
- In material and methods section please to insert the correct catalog number of Col1a-antibody used in WB.
- It was an antibody used in the previous paper [5], and the correct catalog number of Col1a-antibody was entered, but unfortunately, it is not produced anymore.
- In the discussion section, I suggest to improve the final part highlight the important of the data obtained to improve the clinical practice.
-
Line 294-298: From our study, we showed that tendinopathy can be reversed or prevented by inhibition of aberrant differentiation by the stimulation of adequate factors. This was the first study to suggest the possibility that inhibition of the aberrant differentiation of TDSCs could be a therapeutic tool. We hope that our study will be a cornerstone to develop a new regenerative therapeutic agent for tendinopathy.
It is better to check the English language and style. I suggest to check line 130.
I modified the present tense to the past tense.
Line 133-135: Tenomodulin, Scleraxis and Tenascin C were used as tenogenic differentiation markers, and collagen type I and collagen type III were used to confirmed regenerative capacity of TDSCs.

Round 2
Reviewer 2 Report
The authors have done an excellent job, pleasantly expanding the information they had already provided and making the appropriate modifications for a better understanding of their work. I would recommend that you re-edit Figures 2, 3, 4 and 5 assigning the corresponding letters (A, B, C, ...) to the new figures, with their corresponding explanation in the legend.
Author Response
The authors have done an excellent job, pleasantly expanding the information they had already provided and making the appropriate modifications for a better understanding of their work. I would recommend that you re-edit Figures 2, 3, 4 and 5 assigning the corresponding letters (A, B, C, ...) to the new figures, with their corresponding explanation in the legend.
Figures 2, 3, 4 and 5: In accordance with the advice of other reviewers, we re-edit Figures 2, 3, 4 and 5 assigning the corresponding letters, and reflected it in the figure legend and text.
Thank you for your recommendation.
This manuscript is a resubmission of an earlier submission. The following is a list of the peer review reports and author responses from that submission.
Round 1
Reviewer 1 Report
n the manuscript titled “Recovery of Tendon Characteristics by Inhibition of Aberrant Differentiation of Tendon Derived Stem Cells from Degenerative Tendinopathy”, the authors evaluate recovery from aberrant differentiation of tendon derived stem cells (TDSC) after tendinopathyby AGA and T0070907. It was showed that AGA and T0070907 suppressed the abnormal differentiation by results of Alizarin Red S staining and Oil Red O staining. However, the change of differentiation markers seems to be inconsistent with these results. The “Discussion” is very confusing, it's difficult to understand. I also think that this study have other problems as indicated below.
1.It should be described clearly that the cells extracted from 5-week-old and 15-week-old rat models are designated as 5T1-4 and 15T1-4.
The meaning of the abbreviations, 5T1-4 and 15T1-4, should be clearly stated in the text.
TDSG from normal rat should be used as normal control to indicate abnormality of 5T1-4 and 15T1-4 cells.
I understand the decision procedure of the dose of AGA and T0070907. However, I do not see clearly whether the reagents were blocked or not since there were no changes on the protein levels. How about performing the experiments using other reagents or RNAi?
It should be explained that as which cell markers Tenascin-C,Col1,Col3 and α-SMA were used.
5.The authors says “western blot analysis showed no significant differences in the protein levels of collagen I and α-SMA of the 5T-1, 2, and 3 cells between the AGA and control groups. ” in the text (lines 125 to 127). However, there were no “significant difference” marks on 5T-4 AGA in the graphs of “normalized protein levels of Col1” (figure 3B). Please make those consistent.
In the graphs of figure 5, the font size on the x-axis should be changed larger.
I hope these comments will be helpful.
Author Response
Comments and Suggestions for Authors 1
n the manuscript titled “Recovery of Tendon Characteristics by Inhibition of Aberrant Differentiation of Tendon Derived Stem Cells from Degenerative Tendinopathy”, the authors evaluate recovery from aberrant differentiation of tendon derived stem cells (TDSC) after tendinopathy by AGA and T0070907. It was showed that AGA and T0070907 suppressed the abnormal differentiation by results of Alizarin Red S staining and Oil Red O staining. However, the change of differentiation markers seems to be inconsistent with these results. The “Discussion” is very confusing, it's difficult to understand. I also think that this study has other problems as indicated below.
The number in the line is the number applied when the “tracking change” function is used.
It should be described clearly that the cells extracted from 5-week-old and 15-week-old rat models are designated as 5T1-4 and 15T1-4. The meaning of the abbreviations, 5T1-4 and 15T1-4, should be clearly stated in the text.
Line 291-294: The cells of each of the four rats under the condition of 5-week-old, chemically-induced tendinopathy, are labelled as 5T-1, 5T-2, 5T-3 and 5T-4. Similarly, the cells of each of the four rats under the condition of 15-week-old, injury-with-use tendinopathy, are labelled as 15T-1, 15T-2, 15T-3 and 15T-4.
TDSG from normal rat should be used as normal control to indicate abnormality of 5T1-4 and 15T1-4 cells.
Line 91-94: From our previous study, TDSCs extracted from the chemically-induced tendinopathy model were more likely to undergo osteogenic differentiation compared to normal TDSCs, and TDSCs extracted from the injury-with-overuse tendinopathy model were more likely to undergo adipogenic differentiation compared to normal TDSCs [5]. Therefore, the goal of this contribution is to report the therapeutic potential of suppressing aberrant differentiation processes in TDSCs extracted from tendinopathy through the inhibitors AGA and T0070907.
I understand the decision procedure of the dose of AGA and T0070907. However, I do not see clearly whether the reagents were blocked or not since there were no changes on the protein levels. How about performing the experiments using other reagents or RNAi?
Line 264-266: Further study is required to find different inhibitors or to perform the experiments with RNAi that are less cytotoxic and induce tenogenic differentiation and regeneration of TDSCs.
It should be explained that as which cell markers Tenascin-C, Col1, Col3 and α-SMA were used.
Line 121-123: Tenomodulin, Scleraxis, Tenascin C and α-SMA are used as tenogenic differentiatioin markers and collagen I and collagen III are used to confirmed regenerative capacity of TDSCs.
Line 240-245: We confirmed the expression level of the tenogenic marker after treatment of AGA and T0070907. To account for key tendon-related components, this study utilizes a range of tenogenic markers including tenomodulin, scleraxis, tenascin C, α-SMA, collagen I and collagen III [24]. Collagen I, collagen III and tenascin C are components of the extracellular matrix of tendon, and tenomodulin and scleraxis are markers known to be involved in the tenogenic differentiation [25]. In addition, α-SMA is used as an active tenocyte marker in mice and rabbit [26,27].
The authors say “western blot analysis showed no significant differences in the protein levels of collagen I and α-SMA of the 5T-1, 2, and 3 cells between the AGA and control groups.” in the text (lines 125 to 127). However, there were no “significant difference” marks on 5T-4 AGA in the graphs of “normalized protein levels of Col1” (figure 3B). Please make those consistent.
Figure 3B: In accordance with the advice of other reviewers, both alpha1 and alpha2 chains of Col 1 were checked to identify more significant differences. And the graphs of “protein levels of Col1” has been updated normalized protein levels of Col1 with significance bars on 5T-4 AGA.
In the graphs of figure 5, the font size on the x-axis should be changed larger.
To increase the font size, the graphic formats have been modified in all figure.
I hope these comments will be helpful.

Reviewer 2 Report
Recovery of Tendon Characteristics by Inhibition of Aberrant Differentiation of Tendon Derived Stem
Cells from Degenerative Tendinopathy
The authors used tenocytes from tendinopatic tendons and treated them with two inhibitors: 18α-glycyrrhetinic acid (AGA) and T0070907 to block osteogenic differentiation and adipogenic differentiation as a healing in vitro model hypothesizing that inhibition of aberrant cell commitment could alleviate tendinopathy. AGA, but not T0070907 induced tenogenic differentiation in mRNA levels by inhibition of osteogenic differentation. The approach is novel and interesting. However, the manuscript has several draw-backs. A larger panel of key tendon-related components such as mohawk and scleraxis should be included to allow a more reliable statement concerning tenogenic differentiation. The markers measured (tenascin C) are not selective for tenocytes. Another question should be adressed: Would the same treatment with the inhibitors affect non-tendinopathic cells in a similar manner (discuss). The rationale for the investigation of aSMA is not explained also the usage of DMSO to induce adipogenesis and osteogenesis. could it be supported by references? The discussion is short and remains superficial. it should be further elaborated. "This indicates that AGA 180 and T0070907 can prevent tendinopathy by inhibiting this aberrant differentiation" it was only shown at the mRNA level based not based on a conclusive set of markers...the generally high cytotoxicity of the inhibitors should be discussed. What are the side effects expected?
Line 49: Kartogenin implantation, rather „application“?
Line52: „osteocytes are CBFA-1, Runx2, and Osterix“ better „osteoblasts“
Lines 74 and 78: „maximum dose killing“ better to categorize cytotoxic doses.
Figure 1: Labeling of the images „15T1-4“ should be explained in the legend, does it represent 4 days (but in the legend 5 days are mentioned…) or cells of different animals as cell donors?
Line 89: „induce adipogenic differentiation“, better to write „undergo“
Line 87: „chemically-induced tendinopathy model were…“ is this another model then used in the following experiments?
Line 93: „DMSO“ please explain: concentration used (it is cytotoxic) and what ist he rationale to use it?
Line 97: „5T-1 and 5T-4 cells compared to 5T-2 and 5T-3 cells“ the numbering is not explained
Figure 2: DMSO was used at which concentration?
Line 126: alphaSMA, please explain the rationale why was it investigated – it is a myofibroblastic marker
Figure 3B: col 1, range 80-120 kDA, why is this range listed? Was the alpha1 oe alpha2 chain detected?
Why was tenascin C not detected on the protein level? „of tenogenic marker“ neither tenascin C nor aSMA or col1 are tenogenic marker
Line 139: „injury-with-overuse tendinopathy models“ are cells indeed irreversibly changed by tendinopathy or do they recover during passaging in monolayers?
Figure 4: explain why DMSO might induce adipogenesis? % of DMSO?
Legend Figure 5: explain abbreviations (col1, col3, aSMA)
Discussion
The panel of tendon-related markers investigated is not sufficient.
Line 184: „only in a specific cell line“ which cell line? Explain?
Line 187: „AGA blocked gap junctional communication“ discuss this in regard to tendon which has to respond to mechanic impulses by cell-cell communication via gaps. Whih connexins are blocked?
Line 228: „900 mg/kg dose of ofloxacin“ route of application (i.m., p.o.). This model is rather unusual. Why was it selected. The mode of action of fluorquinolones on tendon is still unclear (chelating agents, capturing divalent cations, hampering integrin function…?), effects are often delayed.
Line 241: why was tenomodulin not investigated later as a tendon-related marker?
Line 245-246: which models?
Line 316: calculate „g“ instead of „rpm“
Author Response
Comments and Suggestions for Authors 2
Recovery of Tendon Characteristics by Inhibition of Aberrant Differentiation of Tendon Derived Stem
Cells from Degenerative Tendinopathy
The authors used tenocytes from tendinopatic tendons and treated them with two inhibitors: 18α-glycyrrhetinic acid (AGA) and T0070907 to block osteogenic differentiation and adipogenic differentiation as a healing in vitro model hypothesizing that inhibition of aberrant cell commitment could alleviate tendinopathy. AGA, but not T0070907 induced tenogenic differentiation in mRNA levels by inhibition of osteogenic differentation. The approach is novel and interesting. However, the manuscript has several draw-backs.
The number in the line is the number applied when the “tracking change” function is used.
A larger panel of key tendon-related components such as mohawk and scleraxis should be included to allow a more reliable statement concerning tenogenic differentiation. The markers measured (tenascin C) are not selective for tenocytes.
Line 240-244: We confirmed the expression level of the tenogenic marker after treatment of AGA and T0070907. To account for key tendon-related components, this study utilizes a range of tenogenic markers including tenomodulin, scleraxis, tenascin C, α-SMA, collagen I and collagen III [24]. Collagen I, collagen III and tenascin C are components of the extracellular matrix of tendon, and tenomodulin and scleraxis are markers known to be involved in the tenogenic differentiation [25].
Figure 3, 5: In this study, mohawk was not used as a marker. In line with reviewer feedback, further experimentation was done using scleraxis and tenomodulin and these results have been edited in.
Another question should be adressed: Would the same treatment with the inhibitors affect non-tendinopathic cells in a similar manner (discuss).
Line 03-208: A previous study found evidence that TDSCs extracted from the chemically-induced tendinopathy model were more likely to undergo osteogenic differentiation compared to normal TDSCs, and TDSCs extracted from the injury-with-overuse tendinopathy model were more likely to undergo adipogenic differentiation compared to normal TDSCs [5]. Building upon these findings, this study seeks to investigate the therapeutic potential of suppressing the aberrant differentiation processes in TDSCs from tendinopathy through the inhibitors AGA and T0070907.
The rationale for the investigation of aSMA is not explained also the usage of DMSO to induce adipogenesis and osteogenesis. could it be supported by references?
Line 245: α-SMA is used as an active tenocyte marker protein in mice and rabbit [26,27].
Line 303-306: T0070907 and AGA were dissolved with 100% dimethyl sulfoxide (DMSO, Sigma Aldrich) to make a stock solution and then diluted with Dulbecco’s Modified Eagle’s Medium (DMEM) to a final DMSO concentration of 1%.
Line 317: The control group was treated with DMSO only, and the treatment group was exposed to 25 μM of AGA with DMSO concentrations.
Line 329: The control group was treated with DMSO alone, and the treatment group was exposed to 10 μM of T0070907 with DMSO concentrations.
We did not use DMSO to induce adipogenesis and osteogenesis. We used DMSO to dissolve the inhibitors, so we used the same concentration of DMSO used in the working solution of the drug to replace control. We modified DMSO to control in the Figures to prevent confusion. The rationale can be based on what was used in the previous reference paper (line 463: ref 23).
The discussion is short and remains superficial. it should be further elaborated. "This indicates that AGA 180 and T0070907 can prevent tendinopathy by inhibiting this aberrant differentiation" it was only shown at the mRNA level based not based on a conclusive set of markers...the generally high cytotoxicity of the inhibitors should be discussed. What are the side effects expected?
Both AGA and T0070907 inhibit the aberrant differentiation of TDSCs in Figure 2A, 4A, but these inhibitors decrease only mRNA level of marker in Figure 2B, 4B. However, only AGA induced tenogenic differentiation of TDSCs in mRNA levels in Figure 3A. It seems that these inhibitors are maintained rather than induced tenogenic differentiation and regeneration at protein level except in 5T-4 (Figure 3B, 5B). Further study is required to find different inhibitors that are less cytotoxic and induce tenogenic differentiation and regeneration of TDSCs. It is also important to treat the right amounts of drugs because normal tenocyte can also be killed or altered.
Line 49: Kartogenin implantation, rather „application“?
Line49: We modified it to “application” instead of "implantation”.
Line52: „osteocytes are CBFA-1, Runx2, and Osterix“ better „osteoblasts“
As in the following sentence, it was stated that it was not an osteoblast but an osteocyte in ref [6]. The major transcription factors that have key roles in the differentiation of MSCs into osteocytes are CBFA-1/Runx2 and Osterix.
Lines 74 and 78: „maximum dose killing“ better to categorize cytotoxic doses.
Line 74, 79: We modified it to “cytotoxic dose” instead of " maximum dose”.
Figure 1: Labeling of the images „15T1-4“ should be explained in the legend, does it represent 4 days (but in the legend 5 days are mentioned…) or cells of different animals as cell donors?
Line 82- 85: 5T-1, 5T-2, 5T-3 and 5T-4 stem cells were collected from different rat specimens as 5-week-old, chemically-induced tendinopathy donors. In the same manner, 15T-1, 15T-2, 15T-3 and 15T-4 cells were collected from different rat specimens as 15-week-old, injury-with-overuse tendinopathy donors.
In response to feedback, we have updated Figure 1 (legend/caption) to define 15T-1, 15T-2, 15T-3,and 15-T4 as different rat specimens.
Line 89: „induce adipogenic differentiation“, better to write „undergo“
Line 92, 94: We modified it to “undergo” instead of " induce”.
Line 87: „chemically-induced tendinopathy model were…“ is this another model then used in the following experiments?
We used the same models mentioned in line 97 in the following experiment.
Line 288-294: 5-week-old rats were used to make the chemically-induced tendinopathy model and 15-week-old rats were used to make the injury-with-overuse tendinopathy model. For each condition, four 5-week-old and four 15-week-old rat models were used to extract cells for investigation of osteogenic and adipogenic differentiation, respectively. The cells of each of the four rats under the condition of 5-week-old, chemically-induced tendinopathy, are labelled as 5T-1, 5T-2, 5T-3 and 5T-4. Similarly, the cells of each of the four rats under the condition of 15-week-old, injury-with-use tendinopathy, are labelled as 15T-1, 15T-2, 15T-3 and 15T-4. (methods)
Line 93: „DMSO“ please explain: concentration used (it is cytotoxic) and what is the rationale to use it?
Line 303-306: T0070907 and AGA were dissolved with 100% dimethyl sulfoxide (DMSO, Sigma Aldrich) to make a stock solution and then diluted with Dulbecco’s Modified Eagle’s Medium (DMEM) to a final DMSO concentration of 1%.
We used DMSO to dissolve the inhibitors, so we used the same concentration of DMSO used in the working solution of the drug to replace control. We modified DMSO to control in the Figures to prevent confusion.
The rationale can be based on what was used in the previous reference paper (line465: ref 23).
Line 97: „5T-1 and 5T-4 cells compared to 5T-2 and 5T-3 cells“ the numbering is not explained
The numbering represents cells of different animals as cell donors.
Line 82-85: 5T-1, 5T-2, 5T-3 and 5T-4 stem cells were collected from different rat specimens as 5-week-old, chemically-induced tendinopathy donors. In the same manner, 15T-1, 15T-2, 15T-3 and 15T-4 cells were collected from different rat specimens as 15-week-old, injury-with-overuse tendinopathy donors.
Line 291-294: The cells of each of the four rats under the condition of 5-week-old, chemically-induced tendinopathy, are labelled as 5T-1, 5T-2, 5T-3 and 5T-4. Similarly, the cells of each of the four rats under the condition of 15-week-old, injury-with-use tendinopathy, are labelled as 15T-1, 15T-2, 15T-3 and 15T-4.
Figure 2: DMSO was used at which concentration?
Line 303-306: T0070907 and AGA were dissolved with 100% dimethyl sulfoxide (DMSO, Sigma Aldrich) to make a stock solution and then diluted with Dulbecco’s Modified Eagle’s Medium (DMEM) to a final DMSO concentration of 1%.
We used DMSO to dissolve the inhibitors, so we used the same concentration of DMSO used in the working solution of the drug to replace control. We modified DMSO to control in the Figures to prevent confusion.
Line 126: alphaSMA, please explain the rationale why was it investigated – it is a myofibroblastic marker
Line 245: α-SMA is used as an active tenocyte marker protein in mice and rabbit [26,27].
Figure 3B: col 1, range 80-120 kDA, why is this range listed? Was the alpha1 oe alpha2 chain detected?
Fig 3B, 5B: We omitted range as advised by the reviewer. We modified the figure because alpha1 and alpha2 chains detected together. We cropped down to the lower part of the previous figure and posted a revised graph.
Why was tenascin C not detected on the protein level? „of tenogenic marker“ neither tenascin C nor aSMA or col1 are tenogenic marker
Fig 3, 5: Based on this feedback, we have further identified using the more commonly used tenomodulin and scleraxis on the protein level. Unfortunately, however, scleraxis was not chosen to be used because the level of expression was too low and the background was too severe. There seemed to be little difference in protein expression level of scleraxis between groups.
Line 139: „injury-with-overuse tendinopathy models“ are cells indeed irreversibly changed by tendinopathy or do they recover during passaging in monolayers?
According to our previous study, the histology of tendinopathy model was different from the histology of normal model [5], and the characteristics of cells were different, so we thought that these cells changed by tendinopathy. It also takes a long time to recover, but we extracted cells before going through a recovery period after setting up models due to the injury-with-overuse. As to whether the extracted cells recover over the passage, it is thought that each passage should be tested to determine exactly. But when we experimented with the tendinopathic cells to passage7, there was a difference from normal cells. We think that TDSCs will go through the aging process rather than recover if passage 10 or higher is passed.
Figure 4: explain why DMSO might induce adipogenesis? % of DMSO?
Line 303-306: T0070907 and AGA were dissolved with 100% dimethyl sulfoxide (DMSO, Sigma Aldrich) to make a stock solution and then diluted with Dulbecco’s Modified Eagle’s Medium (DMEM) to a final DMSO concentration of 1%.
Line 329-330: The control group was treated with DMSO alone, and the treatment group was exposed to 10 μM of T0070907 with DMSO concentrations.
We did not use DMSO to induce adipogenesis and osteogenesis. We used DMSO to dissolve the inhibitors, so we used the same concentration of DMSO used in the working solution of the drug to replace control. We modified DMSO to control in the Figures to prevent confusion.
Legend Figure 5: explain abbreviations (col1, col3, aSMA)
Line 404: We explained abbreviations.
Discussion
The panel of tendon-related markers investigated is not sufficient.
Line 184: „only in a specific cell line“ which cell line? Explain?
Line 215, 250 : We modified it to “5T-4” instead of " a specific cell line”.
Line 187: „AGA blocked gap junctional communication“ discuss this in regard to tendon which has to respond to mechanic impulses by cell-cell communication via gaps. Whih connexins are blocked?
Line 218-219: AGA blocked gap junctional communication mediated by connexin 43 in lung cell.
Line 228: „900 mg/kg dose of ofloxacin“ route of application (i.m., p.o.). This model is rather unusual. Why was it selected. The mode of action of fluorquinolones on tendon is still unclear (chelating agents, capturing divalent cations, hampering integrin function…?), effects are often delayed.
We used ofloxacin in reference to the ref [29]. In our previous study, the chemically-induced tendinopathy models were created using this drug and it was confirmed by histology that the tendinopathy was formed [5].
Line 241: why was tenomodulin not investigated later as a tendon-related marker?
Line 240-244: We confirmed the expression level of the tenogenic marker after treatment of AGA and T0070907. To account for key tendon-related components, this study utilizes a range of tenogenic markers including tenomodulin, scleraxis, tenascin C, α-SMA, collagen I and collagen III [24]. Collagen I, collagen III and tenascin C are components of the extracellular matrix of tendon, and tenomodulin and scleraxis are markers known to be involved in the tenogenic differentiation [25].
We replaced tenomodulin with tennacin C because we confirmed it as IF to identify TDSC's character in previous study. However, we took advice and performed additional experiments to supplement the content.
Line 245-246: which models?
Line 288-294: 5-week-old rats were used to make the chemically-induced tendinopathy model and 15-week-old rats were used to make the injury-with-overuse tendinopathy model. For each condition, four 5-week-old and four 15-week-old rat models were used to extract cells for investigation of osteogenic and adipogenic differentiation, respectively. The cells of each of the four rats under the condition of 5-week-old, chemically-induced tendinopathy, are labelled as 5T-1, 5T-2, 5T-3 and 5T-4. Similarly, the cells of each of the four rats under the condition of 15-week-old, injury-with-use tendinopathy, are labelled as 15T-1, 15T-2, 15T-3 and 15T-4.
Line 316: calculate „g“ instead of „rpm“
Line367: the supernatants were obtained by centrifugation at 15,500 g for 10 minutes.
We modified it to “15,500 g” instead of " 13000rpm”

Round 2
Reviewer 1 Report
I think this manuscript would not be suitable for publication in this journal, based on the points I commented before.
Reviewer 2 Report
The authors revised the manuscript thouroughly. Experimental work (Investigation of tendon-related markers) has been added now. Some novel supporting literature has been added.
I have only one remark concerning the fact that alphaSMA is not expressed in vivo by all tenocytes (only a varying and usually a low percentage of cells are positive) and hence, should not be designated as a tenogenic differentiation marker (lines 123 and 238).
typical errors / non-uniform writing
Line 123: please correct: „differentiatioin“
Lines 179-181: tenomodulin and scleraxis has been written in capital letters before. Please use a consistent writing style.
Line 214: adapt font „mediated by connexin 43“
Line 400: Abbreviation list: collagen type I and collagen type III: in the text of the manuscript the authors write alwys only collagen I or collagen III please adapt.